# Anaemia in India and Its Prevalence and Multifactorial Aetiology: A Narrative Review

**DOI:** 10.3390/nu16111673

**Published:** 2024-05-29

**Authors:** D. Ian Givens, Seetha Anitha, Carlotta Giromini

**Affiliations:** 1Institute for Food, Nutrition and Health, University of Reading, Reading RG6 5EU, UK; 2School of Applied Sciences, University of Lilongwe, Area 15, Lilongwe P.O. Box 1614, Malawi; dr.anithaseetha@gmail.com; 3Department of Veterinary Medicine and Animal Sciences, Università degli Studi di Milano, 20122 Milano, Italy; carlotta.giromini@unimi.it

**Keywords:** anaemia, India, iron, vitamin C, vitamin B_12_, folate, millets

## Abstract

The prevalence of anaemia in India remains high in children, especially those in rural areas, and in women of childbearing age, and its impairment of neurological development can have serious lifelong effects. It is concerning that the most recent official data (2019–21) indicate an increased prevalence compared with 2015–16. There is also considerable variability in childhood anaemia between Indian states with socioeconomic factors, such as wealth and education contributing to the risk of anaemia among adolescent women and their children. Dietary iron deficiency is often regarded as the main contributor to anaemia but increasing evidence accumulated from the authors’ ongoing literature database coupled with recent literature research suggests that it has a multifactorial aetiology, some of which is not related to nutrition. This narrative review focused on these multifactorial issues, notably the contribution of vitamin B_12_/folate deficiency, which also has a high prevalence in India. It was also noted that the dietary intake of bioavailable iron remains an important contributor for reducing anaemia, and the role of millets as an improved iron source compared to traditional staple cereals is briefly discussed. The overall conclusion is that anaemia has a multifactorial aetiology requiring multifactorial assessment that must include assessment of vitamin B_12_ status.

## 1. Introduction

The World Health Organisation defines anaemia as the condition of having a low blood haemoglobin concentration, i.e., <130, <120 and <110 g/L for men, non-pregnant women and children aged 6–59 months, respectively [1], although Ghosh et al. [2] have recently suggested that a cut-off value of 110 g/L may be more appropriate for Indian women of childbearing age. WHO [3] reports that, worldwide, 40% of children aged 6–59 months, 37% of pregnant women and 30% of women aged 15–49 years are affected by anaemia. Also, in 2019, anaemia caused 50 million years of healthy life to be lost [4]. A high anaemia prevalence is a major concern in India and many other countries, including some developed countries. It is one of the WHO global targets for reduction, which aims for a 50% reduction in anaemia in women of reproductive age by 2025, although, recently, it was reported that global anaemia prevalence has changed little in the last 20 years and the target may still not be met by 2030 [5].

Iron is an essential nutrient and is a cofactor for many haemoproteins and non-haem iron proteins. It is a component of the haemoprotein haemoglobin in red blood cells and of myoglobin, which, respectively, transport oxygen around the body and store it in muscles and other tissues. A key role of iron is providing the haem iron centre of cytochrome C oxidase, which is essential to its function in the final step of the electron transfer chain that drives ATP synthesis during cellular respiration in mitochondria [6]. There is concern that iron deficiency may compromise this process, which is essential for brain developments [7]. Sub-optimal iron status due to low dietary iron intake is usually regarded as the most common cause of anaemia, especially in low- and middle-income countries (LMICs) [3]. There are, however, increasing concerns that haemoglobin concentration may not be a satisfactory indicator of iron deficiency [8].

As indicated by WHO [3], there are also several non-iron-related factors that can lead to anaemia, including a dietary deficiency of vitamin B_12_ and factors that lead to blood loss from the digestive tract, including worm infestation, which makes the need for differential diagnosis very important. This review focuses on the aetiology of anaemia in India with a focus on the relative contribution from iron deficiency anaemia (IDA) along with measures that have been developed to reduce the prevalence of IDA in children and women of childbearing age.

## 2. Methods

This narrative review is based on a publication database concerning the prevalence of anaemia in India and its various causes and associations built by the authors over a period of about four years. In addition, new publications from 2023 and 2024 were identified by searches using Pub Med (National Library of Medicine).

## 3. Prevalence of Anaemia in India

The prevalence data for anaemia in adults and children as reported by the Indian National Family Health Surveys NFHS-3 (2005–06) [9], NFHS-4 (2015–16) [10] and NFHS-5 (2019–21) [11] are summarised in Table 1. Overall, more than 58% of children aged 6–59 months were anaemic, with haemoglobin (Hb) concentrations of <11 g/dL. Anaemia was more prevalent in children living in rural areas than urban areas, which is mainly related to mothers with little or no education and households of low wealth status. Rural children also suffer more undernutrition and are exposed to poorer sanitary conditions than urban children, which may lead to greater intestinal worm burdens. Also, children’s anaemia status is closely linked to the anaemia status of the mother and early childbearing is most common in rural areas, with the likelihood that these mothers will be more anaemic than older urban mothers [9]. Very similar findings have been seen in Africa, with rural Ethiopian children having a 23% greater risk of anaemia than urban children for similar reasons as seen in India [12].

It is concerning that the lower values seen in the 2015–16 survey compared with 2005–06 increased in 2019–21. The increases relative to 2015–16 were seen in mild (Hb 10.0–10.9 g/dL), moderate (Hb 7.0–9.9 g/dL) and severe anaemia (Hb < 7.0 g/dL), although the prevalence of severe and moderate anaemia remained lower that the values reported in NFHS-2 (1998–99) [13]. There was considerable variability in childhood anaemia between Indian states, with the highest prevalence in Gujarat (79.7%), followed by Jammu and Kashmir (72.7%) and Madhya Pradesh (72.7%), and the lowest seen in Kerala (39.4%) [13]. In addition, the number of states with anaemia prevalence greater than 60% doubled from 5 in 2015–16 to 11 in 2019–21 [13].

Singh et al. [13] reported that the key factors responsible for the increased prevalence in 2019–21 included the low anaemia status of mothers and their socio-economic status and education. They also highlighted the need for more rigorous technology-based monitoring systems for food supplementation programmes in India to reduce the high prevalence of childhood anaemia. The overall prevalence of anaemia in women aged 15–48 years and men was also higher in those residing in rural areas than those in urban environments (Table 1). Anaemia rates for women in 2019–21 were mostly higher than for 2015–16, which supports the link with higher childhood prevalence in that period [13]. There were also concerns that the higher anaemia rates seen in NFHS-5 (2019–21) were related to the use of capillary blood, which has been shown to give lower Hb values than venous blood [14]. The current Diet and Biomarkers Survey in India (DABS-1) will use venous blood to measure Hb [15]

The study of Kumari et al. [16] examined the cross-sectional prevalence of anaemia in 200 adolescent girls in a tertiary care hospital in Bihar in NE India. It found values broadly in agreement with NFHS-4, with 50% of girls aged 10–19 years being anaemic (Hb concentrations <12 g/dL). The study highlighted the substantially increased need for dietary iron for adolescent girls, suggesting that they are particularly susceptible to IDA. A recent study [17] analysed the data for 116,117 and 109,400 adolescent women (aged 15–19 years) obtained from NFHS-4 (2015–16) and NFHS-5 (2019–21), respectively, with the aim of identifying factors that contributed to a rise in adolescent anaemia in the second study compared with the first. In agreement with [13], it was concluded that target interventions for adolescent females were needed, especially in the states with the highest increase in anaemia prevalence. It also highlighted the importance of socioeconomic issues, such as wealth and education, as contributing to the risk of anaemia among adolescent women in India. In addition, it noted that poor nutritional status and adolescent motherhood are key risk factors for adolescent anaemia in this population. The study emphasises the urgency of a determined focused approach to reduce anaemia, particularly in this vulnerable section of the population. The study [17] did reflect on the importance of low dietary intakes of iron, vitamin B_12_ and folate as contributors to anaemia, although it seems that data on the intake of these nutrients were not available for the subjects involved. Such data are clearly urgently needed, especially for female adolescents, but also for all sections of the population.

## 4. The Association of Dietary Iron Intake and Anaemia: Is It Strong?

A knowledge of the contribution of iron deficiency to anaemia is very important for public health since it will inform the development of anaemia reduction programmes and health economists when calculating disability-adjusted life years as a measure of disease burden and hence cost. Whilst IDA is generally regarded as the major contributor to anaemia in India in recent times [18], there is no doubt that its aetiology is multifactorial, with a range of other factors contributing, including other nutritional deficiencies, sickle cell anaemia, thalassaemia, malaria, hookworm infestation and schistosomiasis [19]. Other factors include chronic blood loss from menstruation and gastrointestinal bleeding. In many countries, the proportion of anaemia caused by IDA is unclear since many surveys assess anaemia solely from Hb concentration and do not measure the iron status of the study subjects or their iron intake as noted by [20]. It is notable, however, that, in countries where little haem-iron-rich red meat is habitually consumed, IDA is some six to eight times more prevalent than in North America and Europe [21].

Pasricha et al. [22] examined the origins of anaemia in 401 young children in rural India. This involved a cross-sectional study of children aged 12 to 23 months, in two rural districts of Karnataka. In addition to Hb, a range of important measurements were made, including serum ferritin, folate, and vitamin B_12_, along with malaria infection and hookworm infestation. Anthropometric measurements were made, and nutritional intake was estimated. Overall, anaemia (Hb < 11 g/dL) was seen in 75.3% of the children and was significantly associated with low iron status (low serum ferritin), maternal anaemia and food insecurity. Circulating ferritin was highly associated with iron intake, serum C-reactive protein and maternal Hb. Pasricha et al. [22] concluded that whilst approaches for reducing childhood anaemia must include an optimised iron intake, it must also include maternal anaemia, together with assessments of poverty and food insecurity.

Kassebaum et al. [23] undertook a systematic analysis of the global anaemia burden from 1990 to 2010. Whilst, globally, they found that IDA was the most frequent cause of anaemia, they also highlighted that, in some regions, 10 different conditions were in the top three causes. They also noted that anaemia arising from malaria, schistosomiasis and chronic kidney disease were the only causes to increase in prevalence. Petry et al. [19] carried out a systematic review of national surveys from low, medium, and high Human Development Index countries to assess the proportion of anaemia that was associated with IDA in young children and non-pregnant women of childbearing age. In the 23 countries studied, the contribution of iron deficiency to total anaemia ranged from 0.3 to 50.6% and 2.9 to 63.9% in children and women of childbearing age, respectively. Georgia and Kenya contributed the lowest and highest proportion of iron deficiency to total anaemia, respectively, in both children and women of childbearing age. Petry et al. [19] also showed that, across all countries, the proportion of anaemia that was associated with iron deficiency was only 25% (95% CI: 18–32%) and 37% (95% CI: 28–46%) in the children and women, respectively, notably lower than the 50% previously assumed. The authors claimed that this was the first multi-country study to use a direct measure of iron status, i.e., low serum ferritin concentrations after adjustment for acute inflammation, which may explain the lower associations.

Swaminathan et al. [24] examined the association between iron intake and anaemia in Indian women of reproductive age (15–49 years) using combined data from NFHS-4, 2015–16 and household nutrient and food consumption data from the 68th round survey (2011–12) of the Indian National Sample Survey Office [25]. Overall, they found that the relationship between iron intake and the risk of anaemia was poor (odds ratio of 0.992, 95% CI: 0.991–0.994) with all the factors considered, apart from the length of education, which had a greater impact than iron intake (Table 2).

Being underweight or in the lowest wealth quintile had a much greater effect on increasing the risk of anaemia than iron intake. Further analysis showed increasing iron intake by 1 mg/day reduced the risk of anaemia by only 0.8% after adjusting for a range of potential confounders. The study [24] concluded that the provision of fortified iron alone may not lead to a substantial reduction in anaemia in Indian women of reproductive age and may have variable benefits and risks across Indian states.

A study evaluated the aetiology of IDA at a teaching hospital in Uttar Pradesh in northeast India [26]. This was a cross-sectional study on 102 patients aged 18 to 80 years with anaemia (Hb < 11 g/dL). Each patient was extensively investigated, including complete haematological assessments, stool samples for occult blood, ova, and cysts. In selected patients, in accordance with indications, both upper and lower gastrointestinal (GI) endoscopy, abdominal X-ray and ultrasonography and bone marrow iron staining were carried out. Lower GI endoscopy was performed in 30 patients, resulting in evidence of haemorrhoids (27%) and ulcers in the colon, with subsequent biopsy confirming one case each of carcinoma colon, ulcerative colitis, nonspecific colitis, and nonspecific enteritis. Biopsy during upper GI endoscopy showed chronic duodenitis (2.9%), one carcinoma stomach (0.98%) and one periampullary carcinoma (0.98%). Moreover, about 38% of patients had stools positive for occult blood, and lower GI bleeding was the primary cause of anaemia in 39.2% of patients, followed by upper GI bleeding (26.4%) and menorrhagia (23.4% of females). Hookworm infestation was also a major cause.

This study [26] concluded that, whilst inadequate iron intake was the most common cause of anaemia worldwide, the diagnosis of IDA without the full examination of the underlying aetiology is insufficient and risks missing GI cancers and other serious GI conditions where anaemia may be the only manifestation. Also, in such situations, supplementary iron may appear to improve matters but does not deal with the underlying cause. The need for a much clearer differential diagnosis between IDA and anaemia of chronic disease, which leads to gastrointestinal bleeding, was also highlighted in the meta-analysis of Cotter et al. [27]. The outcome was a summary algorithm for the diagnosis of IDA in subjects with gastrointestinal bleeding. An important recommendation was the use of serum transferrin receptor concentration and/or the ratio of serum transferrin receptor/log ferritin. The paper cited evidence that this ratio enabled an accurate differential diagnosis between IDA and anaemia linked to chronic disease.

A study on the differentiation of the various types of anaemia in 26,765 Indian children aged 1 to 9 years of age and 14,669 adolescents was published in 2020 [28]. The findings are summarised in Table 3. While IDA was responsible for most of the anaemia in the 1–4-year-old group, it only represented 36.5% of total anaemia and was only linked with 15.6% and 21.3% in the 5–9-year-olds and adolescents, respectively. In both older age groups, anaemia due to vitamin B_12_/folate deficiency was responsible for most of the anaemia (~25%), apart from that of other undetermined causes.

Given the high prevalence of vitamin B_12_ deficiency in India [29,30], the substantial presence of vitamin B_12_-related anaemia is perhaps not surprising, but this seems to have had much less attention that IDA. This adds to the evidence for the need to assess vitamin B_12_ and folate status when investigating causes of anaemia and not assuming that most anaemia is IDA. It is notable that the study of Singh et al. [31], which used data from NFHS-5 (2019–21), was unable to include vitamin B_12_ status in their model to predict the determinants of anaemia in rural men because of a lack of data. Consideration should be given to the inclusion of vitamin B_12_ in the laudable weekly iron and folic acid supplementation (WIFS) programme for adolescents (10–19 years of age) that operates across rural and urban areas of India [32]. This study also suggests that children under 9 years of age may also benefit from enhanced vitamin B_12_ status. Another concerning feature of this study was the relatively high (24.5 to 43.6%) proportion of anaemia that was classified as being from other causes. Although serum vitamin B_12_ concentration is the most common method for assessing vitamin B_12_ status, recent work [33] suggests that measuring serum holo-transcobalamin may be a more sensitive biomarker of vitamin B_12_ status in otherwise healthy individuals who consume plant-based diets.

Further recent studies have also highlighted the potential inadequacy of assessing IDA solely based on measuring blood Hb. In a study of uncomplicated pregnancies in two Indian women cohorts assessed in the antenatal and postnatal periods, the relationship between Hb and serum ferritin, a marker of body iron stores, was investigated [34]. They found no evidence of an association between Hb and ferritin, although low ferritin was more common in anaemic women and high ferritin was more common in women with severe anaemia in both cohorts. In the postnatal period, they also examined the association between Hb and soluble transferrin receptors (sTfRs), transferrin saturation (TSAT) and hepcidin and found significant linear associations for all three markers. The overall conclusion was that anaemia in pregnant and postpartum women in India is multifactorial and that Hb alone does not give an adequate assessment of iron deficiency. A recent study of the factors associated with IDA in Ugandan children (6–23 months of age) examined the enhancement of Hb to differentiate IDA from total anaemia by the addition of the red blood cell distribution width (RDW) in the presence of microcytosis or the addition of the Mentzer index (mean cell volume/red blood cell count) [35]. IDA was differentiated from total Hb-based anaemia as Hb < 11 g/dL with RDW >14 in the presence of microcytosis or Hb < 11 g/dL with Mentzer index >13. Thus, 41.5% and 45.3% of total anaemia were assessed as IDA by the Hb + RDW and Hb + Mentzer methods, respectively, with results from the two methods being significantly (*p* < 0.001) correlated. It was suggested that these methods would be valuable tools for IDA differentiation in the absence of the ‘gold standard’ serum ferritin, which, perhaps strangely, was not measured. There may also be a concern about the specificity of RDW, which is also used to assess vitamin B_12_ deficiency and related megaloblastic anaemia [36].

Whilst, overall, it seems that inadequate dietary iron intake and/or absorption is likely to account for a high proportion of anaemia in India, this cannot be assumed. Assessments in addition to Hb to assess iron status (e.g., serum ferritin) and vitamin B_12_ status are needed for differentiation. It is not clear from the information available if vitamin B_12_ status is being measured in the current DABS-1 project [15]. The expanded information on vitamin B_12_ is clearly something that nutritionists devising iron-enriched diets to combat anaemia need to recognise in order to ensure that there is a joint nutrition and medical input to the work to identify the most appropriate remedial approach.

## 5. Iron and Vitamin C: Requirements and Their Dietary Intakes in India

In 2024, an updated report of the nutrient requirements for Indians was published by Indian Council of Medical Research and the National Institute of Nutrition (ICMR-NIN) [37]. For iron, the recommended dietary allowances (RDAs) only differed for children 3–12 months of age, with a value of 6 mg/d compared with 3 mg/d in the earlier 2020 report [38] (Table 4). Daily requirements for iron were prepared in recognition of the widespread prevalence of anaemia, and most evidence pointed to iron deficiency being the main cause. It was noted, however, that despite iron supplementation programmes having been in place for some decades, there has been little reduction in anaemia prevalence and, as described earlier, the latest data show an increase (Table 1). The current iron requirements assume a bioavailability of 8% for adults and adolescents, 9% for infants, 6% for children and 12% for pregnant women, although this will vary considerably between dietary iron sources and the presence of other dietary components.

ICMR-NIN [38], published in 2020, updated RDA values for vitamin C (ascorbic acid), which are compared with the ICMR 2010 values [39] in Table 4 and are generally much higher. For example, for men and non-pregnant women, the RDAs are 80 mg/d and 65 mg/d, respectively, compared with 40 mg/d for both in the ICMR recommendations in 2010. No changes to RDA values for vitamin C were made in the ICMR-NIN summary report in 2024 [37]. Vitamin C intake is important, as it has been known for some time that, among organic acids, vitamin C provides the greatest enhancement of the absorption of non-haem iron [40] This is the result of vitamin C forming a soluble chelate with ferric iron, which prevents the formation of unabsorbable iron compounds. The worthwhileness of providing vitamin C (200 mg every 8 h/d) in conjunction with iron supplements (100 mg every 8 h/d) for adult patients with IDA was challenged in a randomised clinical trial in China [41], although no data on background vitamin C intake were given.

There appears to be relatively few recent extensive studies of iron intake across Indian populations. The results from the Indian National Nutrition Monitoring Bureau (NNMB) [42], which examined the nutritional status of the rural population, showed a median intake across households of 12 mg/d, only about 70% of the quoted RDA of 17 mg/d. Strangely, this RDA was for adult men, with women and older children having higher values (Table 4), which suggests that the RDA value for the household would be higher, with a % below the RDA likely to be higher. There was considerable regional variation, with the three highest median household iron intakes in Madhya Pradesh (19.5 mg/d), Uttar Pradesh (18.8 mg/d) and Gujarat (17.1 mg/d), whilst the lowest intakes were seen in Andhra Pradesh (7.2 mg/d), Tamil Nadu (9.1 mg/d) and Kerala (10.1 mg/d). Moreover, the proportion of households with intakes ≥70% of the RDA was only 52.5%, with the lowest being in Andhra Pradesh at 14.6%. A particular concern was that iron intakes were not even 50% of the RDA in 51–83% of pregnant women. There were also indications of a reduction in iron intakes in recent times. Rammohan et al. [43] found that a large proportion of Indian women subsist on low-iron vegetarian diets and, as a result, were significantly more likely to be iron-deficient than those who were omnivorous. Rammohan et al. [43] recommended, amongst other things, effective public education about iron-rich plant- and animal-derived foods that are effective means of increasing iron intake.

The overall household median intake of vitamin C was 29 mg/d, with the highest value of 57 mg/d recorded in Orissa and the lowest of 16 mg/d in Maharashtra, which was also shown to be associated with a low intake of fruit compared to most other states. Only 50.9% of the population had an intake ≥70% of the RDA (given as 40 mg/d) and 34.9% of the population had a median intake <50% of the RDA. All the %RDA values would be much lower if the updated and higher RDA values for vitamin C (Table 4) were used.

In 2017, the NNMB published a brief report [44] on the diet and nutritional status of the urban population. This reported that, overall, iron intake was 77.6% of the RDA (not specified), although 5–88% of pre-school children had iron intakes less than the RDA. This again reflected considerable variation between states. No daily iron intakes were given but the % meeting the RDA was somewhat higher than in the earlier rural populations study. It was reported that the overall intake of vitamin C in the urban populations exceeded the RDA (not specified) at 128.3% but no details of variation across states were given.

The higher iron and vitamin C intakes in the urban study may reflect a more varied diet, possibly with some red meat contributing to more bioavailable haem iron intake. This was supported by the study of Singh et al. [31] using data from NFHS-5 (2019–21) [11], which showed that 30% of rural India men were anaemic compared with only 20% of urban-dwelling men. Nevertheless, the data available confirm that iron intakes by children and adults appear to be considerably suboptimal, but undertaking regular, more quantitative assessments would help substantially to estimate the contribution of low iron intake to the widespread anaemia.

## 6. Vitamin B_12_ Requirements and Dietary Intakes in India

Table 4 summarises the RDA values for vitamin B_12_ in the ICMR 2010 recommendations [43] and those in ICMR-NIN 2020 [38]. Overall, there was in increase in RDA values in 2020 by a factor of at least two for adults, but with larger increases for children. The requirements were based on the factorial method, which retained the daily excretion of 1 µg/day used in ICMR 2010 [39] and a mean bioavailability of 50%. Using new information on the requirements for foetal growth and vitamin B_12_ output in breast milk for six months, the requirements for both processes were increased. ICMR-NIN, 2024 [37], made essentially no changes to the requirements other than indicating that, for children, the increase in RDA from 1.2 to 2.2 µg/day should occur at 5 instead of 7 years of age.

Neither the NNMB 2012 report [42] nor that in 2017 [44] gave any information on vitamin B_12_ intake. The review of vitamin B_12_ deficiency in India by Sasidharan [29] confirms that, unlike in Western diets, the average Indian mainly vegetarian diet contains no vitamin B_12_. There could also be disease conditions or commonly prescribed drugs, notably metformin and proton pump inhibitors, that would reduce the efficiency of vitamin B_12_ absorption. In addition, it is important to note that Indian diets are also poor sources of folic acid/folates and since these and vitamin B_12_ are metabolically interdependent, each cannot function without the other. It is also a concern that cooking in India can lead to a substantial loss of folates [45].

There is clearly much to be carried out to increase the intake of vitamin B_12_ and folate in India. A recent proposal has suggested the fortification of tea with both nutrients [46].

## 7. Health Consequences of Anaemia

Broadly, anaemia is a condition associated with a reduced number of circulating red blood cells. These contain Hb to transport oxygen around the body, and fewer red blood cells means less Hb and therefore less ability to transport oxygen. This leads to outcomes including tiredness, poor concentration, and shortness of breath. The recent WOMAN-2 trial in Pakistan and parts of Africa has shown that severe anaemia (Hb ≤ 7 g/dL) is strongly associated with postpartum haemorrhage and a seven-times higher risk of death or near miss [47]. The study noted that, worldwide, more than half a billion women of childbearing age are anaemic and, each year, about 70,000 women that give birth die from postpartum haemorrhage. Gora et al. [48] reported that, in India, the occurrence of postpartum haemorrhage is 2–4% following vaginal delivery, although it is 6% following caesarean section, with haemorrhage being responsible for 25% of all maternal deaths. Clearly those women who were already substantially anaemic would suffer the consequences of postpartum haemorrhage to a greater degree than non-anaemic women.

It has been known for some time that, untreated, IDA in infancy is linked with a wide range of neurological disorders, including developmental abnormalities, ischaemic stroke, venous thrombosis, breath-holding and other neurologic problems, including impaired psychomotor development [49]. These effects are possibly due to the impaired myelination of auditory and vagal nerves for which iron is needed for normal myelin synthesis [50]. Iron is essential for normal brain development and neurotransmitter function such that the impact of iron deficiency can be severe, with neurological impairments during gestation and lactation not being reversible. Recent research also highlights that young adults (21 years old) with iron deficiency during infancy, had poor executive control, lower verbal IQ and more frequent inattention and sluggish cognitive tempo symptoms assessed at 10 years old. They also had poorer educational achievement in young adulthood mainly due to inattention [51].

While the pathophysiological consequences of anaemia have been well established, the direct effect of anaemia and/or iron deficiency status on more tangible functional parameters such as occupational performance and work productivity has been less investigated and is less tangible. However, strong evidence that anaemia negatively impacts occupational performance has been reported [52]. Another study evaluated the economic benefit in terms of increased wages in relation to the introduction of Double Fortified Salt resulting in a reduction in anaemia and the relative cost [53]. Men and women with IDA had lower wages (by 25.9%, 95% CI: 11.3, 38.1; and by 3.9%, 95% CI: 0.0, 7.7, respectively) than those without IDA. The authors reported that the introduction of Double Fortified Salt was predicted to reduce the prevalence of IDA (from 10.6% to 0.7% in men and 23.8% to 20.9% in women). The economic benefit–cost ratio of iron fortification at a national level was estimated to be 4.2:1. The benefit-to-cost ratio was predicted to be higher in men, both because of the greater reduction in anaemia and because men are more frequently employed, with higher renumeration. However, the benefit of anaemia reduction goes far beyond wage earning by adults, and, in that sense, the benefit-to-cost ratio can affect school performance in children and adolescents [54], the related fatigue and lack of energy in social interactions and the capacity of mothers to provide care [55]. These results have a strong economic and societal significance in supporting the growing economic and cost-effectiveness assessment of national investment in population-wide iron interventions.

The important contribution of low vitamin B_12_ and folic acid/folate status in India to anaemia prevalence is discussed above, but this low status can have another serious comorbidity: that of an increased risk of neural tube defects (NTDs) developing in the foetus. The birth prevalence of NTDs in India has been estimated at 4.5 per 1000 births [56], with other estimates ranging from 4 to 8 per 1000 births [46], which, overall, is at least 4.5 times higher than in Europe [57] and represents more than 100,000 babies in India each year [46]. The lower prevalence in Europe is no doubt helped by a policy of folic acid supplementation, ideally before conception and for the first 12 weeks of pregnancy, as well as a population with generally adequate vitamin B_12_ status. A policy and provision of folic acid supplementation has not been adopted in India and the low vitamin B_12_ status in women of childbearing age is not fully addressed [46]. It has therefore been proposed that tea, highly consumed in India, would make an excellent vehicle for fortification with folic acid and vitamin B_12_ and would substantially improve the status of some 500 million girls and young women in India. A pilot study has shown that tea is indeed an excellent fortification vehicle, with the potential to reduce/eliminate NTDs and reduce vitamin-B_12_-related anaemia [58].

## 8. Can Millet Consumption Help to Reduce Anaemia?

Millets have been a traditional staple food in India for some 3000 years, although this has declined since the 1960s due to increased replacement by more affordable refined wheat, maize, and rice. Millets have the benefits of climate and drought resilience and are considerably more nutrient-dense than refined wheat and refined rice [59]. Some millets have a considerably higher iron concentration than maize, refined wheat, and refined rice, which, according to the Indian Food Composition Tables 2017 [60], typically contain 2.49, 1.77 and 0.65 mg iron/100 g, respectively. Pearl and finger millets in the same tables contain 6.42 and 4.62 mg iron/100 g, respectively, but recent studies have extended knowledge on the iron content of millets. A selection is given in Table 5 that shows considerable variability both within and between millet types.

There are also some values in Table 5 that closely agree between data sources, suggesting a common primary source. Overall, pearl millets tend to have the highest iron content, although there is considerable variability and, critically, a high iron concentration may not reflect the greater amount of absorbed iron since its iron bioavailability will be a major contributor to the amount absorbed. No information on iron bioavailability was provided in any of the papers and is clearly a priority for research.

A recent systematic review and meta-analysis [66] assessed whether millet (pearl, finger, sorghum) consumption contributed to increasing blood haemoglobin compared with refined rice and/or wheat. Twelve papers with thirteen eligible studies were included. They showed high heterogeneity (*I*^2^ = 80%) but, using a random effects model, the standardised mean difference (0.68: 95% confidence interval 0.33–1.02; *p* < 0.01) indicated a significant benefit to the millet-containing diets. Overall, compared to refined rice/wheat diets, the millet diets increased haemoglobin by 8.8% (*p* < 0.05), equivalent to 0.90 g/dL. It is interesting to note that sub-group analysis showed that the positive treatment effects were only seen in the studies with children, possibly related to having more studies than with adults and expressing lower heterogeneity. Regrettably, none of the studies in the meta-analysis reported iron intakes and other limitations existed, but they give some confidence that, with further research, millets will have the potential to enhance iron intake and absorption, leading to a reduction in IDA, although, as discussed, iron may not be the only limiting factor and, clearly, millets will not provide any vitamin B_12_.

The results from a one-year dietary intervention study involving pre-school children (3–6 years of age) in Telangana State, India were recently published [67]. The treated children (*n* = 306) were fed six nutrient-rich formulations of millet–pulse–groundnut-based food products ranging in iron concentration from 1.9 to 5.6 mg/100 g, whilst the control children (*n* = 81) did not receive these foods. Estimates of iron intakes in both groups were not reported but blood Hb concentration in the treatment group increased from 9.70 ± 0.14 g/dL to 11.08 ± 0.13 g/dL (mean increase of 1.38 g/dL, *p* < 0.0001), which meant that those initially with moderate anaemia (Hb 7.0–9.9 g/dL) moved into the normal Hb category. Hb in the control group also increased (*p* < 0.0001) but those initially in the moderate anaemia category moved only to the mildly anaemic group, although this may have been a result of the lower baseline Hb values in the control group than in the treatment group. More intervention studies are needed in children and young women.

## 9. Conclusions

The prevalence of anaemia in India remains high in children, especially those in rural areas, and in women of childbearing age. It is concerning that the most recent official data (2019–21) indicate an increased prevalence compared with the period of 2015–16. There is also considerable variability in childhood anaemia between Indian states and with socioeconomic factors, such as wealth and education contributing to the risk of anaemia among adolescent women. There is now no doubt that anaemia has a multifactorial aetiology. As shown in one study [28] with children and adolescents, whilst iron deficiency was responsible for most of the anaemia in the 1–4-year-old group, it only represented 36.5% of total anaemia and only 15.6–21.3% in the older children and adolescents. Crucially, in both older age groups, vitamin B_12_/folate deficiency was responsible for most of the anaemia (~25%), apart from that of other undetermined causes. The consequences of anaemia are very substantial, impacting the neurological development of children, leading to poorer educational attainment and, later, reduced occupational performance, productivity, and income. The impact of anaemia in young women is perhaps less discussed but is associated with greater postpartum haemorrhage and a consequential increased mortality risk. Moreover, a low vitamin B_12_ and/or folate status increases the risk of foetal neural tube defects, which is already 4.5 times higher than in Europe. Improving vitamin B_12_/folate status in women of childbearing age should be a priority and the proposed fortification of tea would seem to be an important option. The potential of high-iron millets to reduce IDA looks promising although work on assessing the bioavailability of the iron is urgently needed and obviously these millets will not solve the widespread vitamin B_12_ deficiency, which also needs urgent attention. Overall, the available evidence points to anaemia having a multifactorial aetiology requiring a multifactorial assessment. Future reviews, including systematic reviews, should begin by acknowledging that anaemia in India and probably some other countries does indeed have a multifactorial aetiology. This should be enshrined in the search criteria to ensure that all contributory factors are assessed.

## Figures and Tables

**Table 1 nutrients-16-01673-t001:** Prevalence of anaemia in India 2005–2021.

Population	NFHS Report ^1^	% Anaemic in Residence		
		Urban	Rural	All
Children 6–59 months	3: 2005–06	63.0	71.5	69.5
	4: 2015–16	56.0	59.5	58.6
	5: 2019–21	64.2	68.3	67.1
Women 15–48 years not pregnant	3: 2005–06	NG ^2^	NG	53.2
	4: 2015–16	51.0	54.4	53.2
	5: 2019–21	54.1	58.7	57.2
Women 15–48 years pregnant	3: 2005–06	NG	NG	58.7
	4: 2015–16	45.8	52.2	50.4
	5: 2019–21	45.7	54.3	52.2
Men 15–49 years	3: 2005–06	NG	NG	24.2
	4: 2015–16	18.5	25.3	22.7
	5: 2019–21	20.4	27.4	25.0

^1^ Definition of anaemia in NFHS 3 [9]: 2005–06 Hb children < 11.0 g/dL, women < 12; men < 13; NFHS 4 [10]: 2015–16 Hb children < 11.0 g/dL, non-pregnant women < 12, pregnant women < 11, men < 13; NFHS 5 [11]: 2019–21 Hb children < 11.0 g/dL; non-pregnant women < 12; pregnant women < 11; men < 13. ^2^ Not given.

**Table 2 nutrients-16-01673-t002:** Factors associated with anaemia in Indian women of childbearing age [24].

Factors ^1^	OR ^2^ for Association with Anaemia (95% CI ^3^)
Underweight vs. Normal	1.216 (1.196–1.237)
Poorest vs. Richest	1.170 (1.134–1.207)
Poorer vs. Richest	1.096 (1.067–1.126)
Middle income vs. Richest	1.066 (1.042–1.091)
Richer vs. Richest	1.043 (1.023–1.065)
Household size > 4 vs. 0–4	1.032 (1.018–1.046)
Other drinking water vs. Piped	1.029 (1.012–1.045)
Open defecation vs. Toilet	1.018 (1.000–1.037)
Iron intake (mg/d)	0.992 (0.991–0.994)
Years of education	0.989 (0.988–0.991)
Overweight vs. Normal	0.787 (0.773–0.801)
Obese vs. Normal	0.785 (0.762–0.808)

^1^ For details, see the original paper. ^2^ Odds ratio derived from model including all the listed factors. ^3^ Confidence interval.

**Table 3 nutrients-16-01673-t003:** Characterisation of the types of anaemia prevalent among children and adolescents aged 1–19 years in India from [28].

	Anaemia Prevalence (%)	Type of Anaemia (% of Total Anaemia)				
Age Group		Iron Deficiency	Folate or Vitamin B_12_	Dimorphic	Inflammation	Other Causes
1–4 years	40.5	36.5	18.9	13.5	6.5	24.5
5–9 years	23.4	15.6	24.6	10.7	5.4	43.6
Adolescents	28.4	21.3	25.6	18.2	3.4	31.4

**Table 4 nutrients-16-01673-t004:** Recommended dietary allowances (RDAs) for vitamin B_12_, iron and vitamin C for Indians.

		ICMR (2010) [39]			ICMR-NIN (2020) [38]		
Sex/Age	Category	Vitamin B_12_ (µg/d)	Iron (mg/d)	Vitamin C (mg/d)	Vitamin B_12_ (µg/d)	Iron (mg/d)	Vitamin C (mg/d)
Men		1.0	17	40	2.2	19	80
Women	NPNL ^1^	1.0	21	40	2.2	29	65
	Pregnant	1.2	35	60	2.45	27	80
	Lactating	1.5	25	80	3.2	23	115
Infants	6–12 month	0.2	5	25	1.2	3(6 ^2^)	30
Children	1–3 year	0.2	9	40	1.2	8	30
	4–6 year	0.2–1.0	13	40	1.2–2.2 ^3^	11	35
	7–9 year	0.2–1.0	16	40	2.2	15	45
Boys	10–12 year	0.2–1.0	21	40	2.2	16	55
Girls		0.2–1.0	27	40	2.2	28	50
Boys	13–15 year	0.2–1.0	32	40	2.2	22	70
Girls		0.2–1.0	27	40	2.2	30	65
Boys	16–18 year	0.2–1.0	28	40	2.2	26	85
Girls		1.0	26	40	2.2	32	70

^1^ Not pregnant, not lactating; ^2^ Adjusted value ICMR-NIN 2024; ^3^ ICMR-NIN 2024 recommends that 2.2 µg/d starts at 5 year.

**Table 5 nutrients-16-01673-t005:** Variation in the iron concentration within and between millet types (mean values or range reported).

Millet Type	Iron (mg/100 g)	Millet Type	Iron (mg/100 g)
Anitha et al. [61]		Sabuz et al. [62]	
Pearl D ^1^	8.5	Pearl	8.3
Pearl P	4.7	Foxtail	2.8
Finger VR847	2.9		
Finger GPU28	2.6		
Nithiyanantham et al. [63]		Chaurasia and Anichari [64]	
Pearl	7.5–16.9	Pearl	11.0
Finger	3.9–7.5	Finger	3.9
Foxtail	2.8–19.0	Foxtail	2.8
Proso	0.8–5.2	Kodo	1.7
Kodo	0.5–3.6	Little	9.3
Little	9.3–20.0	Sorghum	5.4
Barnyard	15.2	Rice ^2^	1.8
		Wheat ^2^	3.5
Ambati and Sucharitha [65]			
Pearl	16.5		
Barnyard	15.2		
Sorghum	3.4		
Finger	3.9		
Kodo	0.5		
Little	9.3		
Proso	0.8		

^1^ D, Dhanshakthi; P, Proagro9444; ^2^ For comparison.

## Data Availability

No new data were created or analysed in this study. Data sharing is not applicable to this article.

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
