# Peer review of "Anaemia in India and Its Prevalence and Multifactorial Aetiology: A Narrative Review"

_nutrients, 2024, doi:10.3390/nu16111673_

Round 1

Reviewer 1 Report

Comments and Suggestions for Authors

This review article is submitted to the "Nutrition and Obesity" section within the Special Issue titled "Diet, Inflammation, and Metabolic Complications." It presents a narrative review focusing on the etiology of anemia in India, with a specific emphasis on the contribution of iron deficiency anemia (IDA) and the measures developed to mitigate its prevalence among children and women of childbearing age.

The article is structured into several sections: prevalence of anemia in India, the association between dietary iron intake and anemia, iron and vitamin C requirements and dietary intakes in India, vitamin B12 requirements and dietary intakes in India, health consequences of anemia, and the potential role of millet consumption in reducing anemia. This organized structure enhances comprehension of this complex and multifactorial topic.

The conclusions drawn from this review are based on the identified contributions of the literature reviewed. The bibliography is robust and the integration of information is comprehensive.

Commentary: However, I believe this topic warrants further investigation due to its complexity, indicating ongoing work is necessary. Therefore, I recommend clarifying the methodology employed to facilitate connections with future reviews.

Reviewer 2 Report

Comments and Suggestions for Authors

This is a quite interesting review article with quite novelty. Howevers, several points should be addressed.

- Subheadings should be added in the Abstract of the article (e.g. Background, Methods, Results, and Conclusion) based on the guidelines of the journal.

- The authors should clearly report in the Abstract the methods used to collect the data that are included in their manuscript.

- The Introduction section is too short without reporting the literature gap that the present review study will cover.

- The 2nd paragraph of the Introduction should be increased by adding more information concerning the functions of Fe in human body, e.g. as a cofactor, etc.

- A separate Methods section should be added after the Introduction section, reporting the methodology used to collect all the article included in the presen manuscript.

- In section 2, line 57, the authors should provide some evidence why anemia in children is more frequently in thos living in rural than urban areas in onjuction with relevant reference. Are any data for other countries in this topic? Please report if they are, providing more information with relevant references.

- The text in lines 89-98 need some relevant references.

- The sentence in lines 113-115 " In many countries the proportion of anaemia caused by IDA is unclear since many surveys that assess anaemia solely from Hb concentration and do not measure the iron status of the study subjects or their iron intake. " need relevant references.

- Are there any data for India concerning whether iron, folic acid and vitamin B12 are prescribed during the period of pregnancy?

- Are there any data for India concerning the most frequent nutritional habits of the general population of India. If meat and meat products, as well as daity products are scarely consumed, then these are potential risk factors for anemia. The authors should use relevant data from the international literature.

Comments on the Quality of English Language

Minor editing of English language required

Round 2

Reviewer 2 Report

Comments and Suggestions for Authors

The authors have significantly improved their manuscript after revision process.